# LEDGF/p75 Is Required for an Efficient DNA Damage Response

**DOI:** 10.3390/ijms22115866

**Published:** 2021-05-30

**Authors:** Victoria Liedtke, Christian Schröder, Dirk Roggenbuck, Romano Weiss, Ralf Stohwasser, Peter Schierack, Stefan Rödiger, Lysann Schenk

**Affiliations:** 1Faculty of Natural Sciences, Brandenburg University of Technology Cottbus-Senftenberg, 01968 Senftenberg, Germany; Victoria.Liedtke@b-tu.de (V.L.); Ch.Schroeder@b-tu.de (C.S.); Dirk.Roggenbuck@b-tu.de (D.R.); Romano.Weiss@b-tu.de (R.W.); Ralf.Stohwasser@b-tu.de (R.S.); Peter.Schierack@b-tu.de (P.S.); Stefan.Roediger@b-tu.de (S.R.); 2Faculty of Health Sciences Brandenburg, Brandenburg University of Technology Cottbus-Senftenberg, 01968 Senftenberg, Germany

**Keywords:** LEDGF, CRISPR/Cas9, DNA damage signaling, γH2AX, ubiquitination

## Abstract

Lens epithelium-derived growth factor splice variant of 75 kDa (LEDGF/p75) plays an important role in cancer, but its DNA-damage repair (DDR)-related implications are still not completely understood. Different LEDGF model cell lines were generated: a complete knock-out of LEDGF (KO) and re-expression of LEDGF/p75 or LEDGF/p52 using CRISPR/Cas9 technology. Their proliferation and migration capacity as well as their chemosensitivity were determined, which was followed by investigation of the DDR signaling pathways by Western blot and immunofluorescence. LEDGF-deficient cells exhibited a decreased proliferation and migration as well as an increased sensitivity toward etoposide. Moreover, LEDGF-depleted cells showed a significant reduction in the recruitment of downstream DDR-related proteins such as replication protein A 32 kDa subunit (RPA32) after exposure to etoposide. The re-expression of LEDGF/p75 rescued all knock-out effects. Surprisingly, untreated LEDGF KO cells showed an increased amount of DNA fragmentation combined with an increased formation of γH2AX and BRCA1. In contrast, the protein levels of ubiquitin-conjugating enzyme UBC13 and nuclear proteasome activator PA28γ were substantially reduced upon LEDGF KO. This study provides for the first time an insight that LEDGF is not only involved in the recruitment of CtIP but has also an effect on the ubiquitin-dependent regulation of DDR signaling molecules and highlights the role of LEDGF/p75 in homology-directed DNA repair.

## 1. Introduction

Dense fine speckled autoantigen of 70 kDa (DFS70) also known as lens epithelium-derived growth factor (LEDGF), PSIP1 (PC4 and SFRS1 interacting protein 1), or transcriptional co-activator p75 is considered a ubiquitous nuclear transcription co-activator [1]. It is linked to various diseases such as cancer and acquired immunodeficiency syndrome (AIDS), and diverse inflammatory conditions have been described as well [2,3]. LEDGF/p75 shows anti-apoptotic activity by promoting the repair of DNA double-strand breaks (DSBs) via the homology-directed repair pathway (HDR) [4], and its overexpression in different cancer cell lines and solid tumors has been linked to tumor progression, aggressiveness, and chemoresistance [5]. In contrast, the shorter splice variant LEDGF/p52 has been proposed to play a pro-apoptotic role and appears to be involved in RNA-splicing [6,7]. LEDGF/p75 is a multi-functional, chromatin-binding protein upregulated in different solid cancer and cancer cell lines [1], promoting the activation of pathways involved in proliferation, cell survival, and DNA repair [4,8]. The SUMOylation of LEDGF/p75 [9] by sumo-specific protease-1 regulates its binding in promoter regions of stress-related proteins. This protects cells from stress-induced necrosis and enhances the activation of the Akt/ERK signaling pathway, resulting in an increased tumor aggressiveness [1,10]. In addition to the upregulation in response to oxidative stress, the mechanism by which LEDGF/p75 protects cancer cells from stress-induced necrosis is not clarified. However, LEDGF/p75 can activate the expression of cancer-related genes e.g., vascular endothelial growth factor C (VEGF-C) as a transcriptional coactivator by protein–protein interaction [4,11,12]. It has been shown in prostate cancer cells that LEDGF/p75 facilitates chemotherapy resistance by counteracting caspase-independent apoptosis [12]. DNA damage response (DDR) is a finely tuned signaling network required to repair potentially lethal DNA double-strand breaks (DSBs) and other DNA lesions. LEDGF/p75 interacting with this network supports rapid repair of DSBs. Thus, LEDGF/p75 recruits the histone acetyltransferase KAT5 to the chromatin, which acetylates histone 4 (H4) at lysine K16 [13]. This acts as a switch between HDR and non-homologous end-joining (NHEJ). Upon H4 acetylation, the BRCA1–BARD1 complex can bind to DNA, supporting HDR. Without H4 acetylation, BRCA1–BARD1 binding is inhibited by 53BP1, which in turn triggers NHEJ [14]. Interestingly, BRCA1 acts upstream (DNA damage sensor) and downstream (participation in various DNA repair complexes) of LEDGF [15]. The termination of DDR is accomplished by the ubiquitination of key regulators such as γH2AX or BRCA1 [16,17]. The LEDGF-dependent BRCA1–BARD1 complex is a member of the E3-ubiquitin protein ligase family [18]. It is involved in ubiquitin-dependent regulation and signal termination of γH2AX [19]. Ubiquitination triggers degradation by the ubiquitin–proteasome system (UPS) and ubiquitin-independent proteasome pathway (UIPP) [20].

We have created LEDGF knockout (KO), EGFP-LEDGF/p75 re-expressing (LEDGF/p75 o/e), and mEmarald_LEDGF/p52 re-expressing (LEDGF/p52 o/e) cells using CRISPR/Cas9 technology to analyze the expression of various DDR proteins. Our work showed for the first time the participation of LEDGF/p75 in the ubiquitin-dependent regulation of DDR signaling molecules.

## 2. Results

### 2.1. CRISPR/Cas9-Generated LEDGF Cell Models

For complete knock-out of the LEDGF gene *PSIP1*, the sgRNA was designed within exon 1 of the *PSIP1* gene to target all splice variants (Figure 1). Then, HEp-2 WT and U2OS WT cells were transfected with non-viral px458_DFS70_E1 vector co-expressing EGFP as a marker and Cas9 enzyme and enriched via EGFP-directed FACS sorting (Appendix A) following single cell out-growth. The LEDGF KO HEp-2 clones were verified at a protein and genomic level (Figure 1E and Appendix A). Potential genomic off-target loci were checked by sequencing and exhibited all unmodified loci (Appendix A). The reconstitution of LEDGF in LEDGF KO was realized by the integration of either EGFP-LEDGF/p75 expression cassette (Figure 1B) or mEmarald_LEDGF/p52 expression cassette at the human safe harbor locus (AAVS1) (Figure 1E,F and Appendix A). EGFP-LEDGF/p75 and mEmarald_LEDGF/p52 incorporation and constitutive expression was confirmed by detecting the fluorescent LEDGF fusion protein (Figure 1C). Both expressed splice variants showed the typical nuclear localization. Additionally, C-terminal LEDGF antibody was used to detect the wild-type LEDGF and EGFP-LEDGF/p75 to show the typical dense fine speckled nuclear staining pattern (Figure 1D). Note, mEmarald_LEDGF/p52 cannot be detected with this antibody, as p52 is missing the C-terminus.

### 2.2. Depletion of LEDGF Decreases Cellular Migration

LEDGF has been previously shown to affect cell migration. Therefore, the cell migration of HEp-2 and U2OS cells was checked. Indeed, the migratory capacity was significantly reduced upon LEDGF knockout in HEp-2 and U2OS cells (Figure 2). LEDGF/p52 re-expression failed to restore the migration capacity of the HEp-2 WT (Figure 2A,C). In contrast, LEDGF/p75 re-expression (WT level) reversed the inhibiting effect, and the cell migration ability was further improved with higher LEDGF/p75 levels (oe) in comparison to the unmodified WT cells (Figure 2C and Appendix A). Additionally, EGFP-LEDGF/p75 o/e cells showed a changed morphology toward an elongated, fibroblast-like phenotype in combination (Figure 2D) with an increased expression of the cytoskeleton subunit α-tubulin (Figure 2B). Morphological analysis revealed that LEDGF/p75 o/e cells exhibited a significantly increased eccentricity and a significantly decreased round shape by 50% (*p* < 0.05, Figure 2D). However, LEDGF KO cells showed a reduced expression of α-tubulin but no change in morphology (Figure 2B).

### 2.3. LEDGF Depletion Sensitizes Cancer Cells toward Etoposide

Already as a result of LEDGF KO, cells showed a significant decrease in cell proliferation in comparison to the WT cells, while no changes in the cell cycle were detectable (Figure 3A and Appendix A). The re-expression of LEDGF/p75 completely rescued the proliferation rate, implicating that the p75 splice variant is the most relevant for cell growth. For the proliferation analysis, the cell lines were used within 10 passages. During this time, no changes in growth behavior could be detected. As shown in Figure 3A, the lower proliferation rate was reflected by a prolonged doubling time (WT 21 h and KO 26.5 h). In addition, LEDGF KO cells were passaged 30 times without showing any visible increased apoptotic phenotype nor complete death. Upon etoposide exposure, LEDGF KO cells showed a significantly reduced cell survival compared to the WT (Figure 3B). Furthermore, the recovery rate after etoposide withdrawal was lower in LEDGF KO cells, as confirmed by ED_50_ determinations suggesting that LEDGF plays a critical role in chemosensitivity toward etoposide (Figure 3C). We were able to compensate the reduced survival of LEDGF KO cells (ED_50_ = 0.123 ± 0.01) by re-expressing LEDGF (EGFP-LEDGF/p75 o/e, ED_50_ = 0.4 ± 0.04), which resulted in an etoposide resistance equal to unmodified WT-cells (ED_50_ = 0.317 ± 0.03) (Figure 3B,C).

### 2.4. LEDGF Depletion Impairs DNA Damage Response via Homology-Directed Repair

LEDGF has been previously implicated to play a crucial role in HDR-mediated damage response involving CtIP-BRCA1-RPA32 signaling [21]. Therefore, we investigated RPA32 foci formation combined with yH2AX foci and DNA fragmentation to detect active HR.

LEDGF knockout in HEp-2 cells caused a significant increase in sensitivity toward etoposide shown by a higher killing rate (Figure 3B). This phenotype was underlined by phosphorylation of H2AX (γH2AX, Figure 4) and an increased level of DNA fragmentation (Figure 4B) indicating etoposide-induced DSBs; however, it was similar to wild-type HEp-2 cells. In contrast, the phosphorylation of RPA32 (pRPA upper band in Figure 4A after etoposide treatment) was almost completely abolished in LEDGF KO cells, while wild-type cells showed an elevated phosphorylation level. This was also reflected in significantly reduced RPA32 foci formation (Figure 4D,E) after etoposide treatment in LEDGF KO cells, indicating an inhibition of CtIP-BRCA1-mediated homology-directed repair. The re-expression of LEDGF/p75 was able to rescue the RPA32 foci formation (Figure 4D,E).

### 2.5. LEDGF Depletion Results in Dysfunctional DNA Damage Response

Interestingly, γH2AX was already increased without DSB-inducing agents in LEDGF KO cells (Figure 4A). Additionally, pulse-field electrophoresis revealed a significantly increased amount of DNA fragmentation in untreated LEDGF KO cells (Figure 4B,C). Surprisingly, LEDGF KO cells exhibited not only a significantly higher number of persistent γH2AX foci (Figure 5 and Appendix A) but also a significantly increased foci formation (foci/nucleus are shown in Appendix A) of the DNA damage response molecule BRCA1 (Figure 5A,B and Appendix A). Both elevated foci formations were reversed by the re-expression of EGFP-LEDGF/p75 (Figure 5A–D and Appendix A).

Despite the high “basal” γH2AX foci in LEDGF KO cells, indicating permanent DNA damage, the cells were able to be maintained over 20–30 passages without dying even though at a significantly slower growth rate (Figure 3A and data not shown). Therefore, we were interested in whether the sustained γH2AX foci are related to a dysregulated degradation of the histone. It has been previously stated that ubiquitination plays an important role in the regulation of DNA damage response signaling. For the γH2AX molecule, K63-linked ubiquitination by UBC13 is important in order to activate the BRCA1-A complex, which coordinates the release of γH2AX followed by subsequent degradation of the proteasome (as illustrated in Figure 5G [16]). This happens downstream of LEDGF-induced CtIP-BRCA1-RPA32 DNA damage signaling.

Here, we show for the first time that LEDGF depletion had a significantly reducing effect on the protein expression levels of UBC13 as well as PA28γ also known as REGy (Figure 5E in Hep-2 and Figure 5F in U2OS cells), which was reversed by LEDGF/p75 re-expression but not LEDGF/p52. This indicates a direct or an indirect role of LEDGF/p75 on the expression of UBC13 and PA28γ, which in turn might affect γH2AX and BRCA1 degradation. In Figure 5G, we propose how LEDGF might regulate HDR-mediated DNA repair in addition to the recruitment of CtIP due to the changing UBC13 and PA28γ levels.

## 3. Discussion

LEDGF/p75 has been reported to be overexpressed in different solid tumors and cancer cell lines [10]. Particularly, it is involved in cancer progression by controlling the expression of genes regulating the cell cycle, cell proliferation, and survival [22]. Furthermore, LEDGF/p75 is supposed to enhance HDR by promoting CtIP–BRCA1-dependent DNA end-resection after DNA DSB and influences the recruitment of DNA-damage response-related downstream proteins such as RPA32 [21]. Instead of siRNA knockdown with potential residual LEDGF expression, we generated complete *PSIP1* knockout using CRISPR/Cas9 technology. We show for the first time that complete LEDGF depletion has an essential influence on DDR signaling.

Firstly, we generated LEDGF KO cell lines. To avoid off-target effects due to plasmid integration and constitutive active Cas9 expression, LEDGF knockout generation was pursued using EGFP reporter expression to enrich cells transiently expressing Cas9 and sgRNA. Thus, effects detectable for EGFP-selected LEDGF KO clones should be based on LEDGF knockout, making the results more reliable. Potential off-target effects due to the sgRNA [23] were checked by sequencing (Appendix A). Additionally, different LEDGF KO clones were tested for their uniform behavior using a proliferation assay (Appendix A). Moreover, LEDGF recovery experiments showed that the effects induced by LEDGF depletion could be rescued by EGFP-LEDGF/p75 re-expression, indicating a specific knockout.

Increased LEDGF/p75 expression in prostate cancer [24], breast cancer [22], or colon cancer [1] was linked with an aggressive tumor phenotype. Furthermore, the upregulation of LEDGF in prostate and breast cancer cell lines has been shown to play a role in proliferation, migration, and chemoresistance [12,22]. Migration and invasion is an important step in cancer metastasis [25]; however, the involvement of LEDGF in this dynamic process remains unclear. In the presented study, proliferation and migration analysis of LEDGF depleted cells (Figure 2 and Figure 3) showed a significant reduction in cell growth and decreased migration ability, supporting LEDGF’s involvement in pro-survival pathways as previously described [12]. This is further underlined by the LEDGF/p75 o/e model in our study where increased expression resulted in enhanced migratory abilities (Figure 2C). Moreover, these cells showed a change in morphology to a fibroblast-like phenotype, which was accompanied with an increased expression of α-tubulin. Up to now, only acetylation of α-tubulin has been connected to cancer cell migration and invasion [26,27]. However, increased class III ß-tubulin has been associated with a more aggressive tumor phenotype [28]. Taking into account both the increased migration capacity and the changed morphology in consideration, we suggest that LEDGF o/e leads to a more aggressive and invasive cancer phenotype.

The role of LEDGF in DDR signaling should be investigated. Therefore, we induced DNA DSB using the topoisomerase etoposide. Interestingly, LEDGF depletion sensitizes cells toward caspase-dependent cell death on etoposide exposure [12]. In this work, the characterization of LEDGF KO clones confirmed a direct correlation of LEDGF depletion with a reduced proliferation and an increased sensitivity toward topoisomerase II inhibition by etoposide.

As previously shown elsewhere [6,21,29], the re-expression of the shorter splice variant LEDGF/p52 has no enhancing effect on the cell proliferation, migration, or chemosensitivity.

Daugaard et al. [21] demonstrated LEDGF as an important factor in DNA repair via HDR. After induction of a DNA DSB, the ensuing cellular damage response leads either to NHEJ or HDR. In contrast to NHEJ, HDR is only active in late S- and G2-phase [30]. Moreover, LEDGF KO cells exhibit a decreased survival and elevated DNA fragmentation upon etoposide exposure, which might indicate a deficiency to repair DNA DSBs by HDR. Therefore, RPA32 foci formation was investigated, and indeed, in LEDGF knockout clones, less RPA32 was recruited to the DNA damage sites in comparison to the wild-type cells. In fact, LEDGF/p75 is known to bind to methylated histones, supporting the binding of C-terminal binding protein (CtBP) interacting protein (CtIP), which assists DNA damage recognition by the MRN complex [21]. Subsequently, this activates three main damage response-related protein kinases: ATM, ATR, and DNA PKs. RPA32, also active during replication, is activated by ATM and ATR, resulting in phosphorylated RPA32 foci formation, which is necessary for the recruitment of downstream DNA repair proteins [31].

The reduced amount of RPA32 foci in LEDGF-KO clones is consistent with the results of Daugaard et al. [21], which implies that LEDGF is necessary for the recruitment of HDR-related DNA repair proteins and is required for an efficient DNA repair. In addition to the drug-induced DNA damage, elevated γH2AX foci formation was already detected upon LEDGF KO, suggesting that LEDGF plays a role in the maintenance of genome stability. In fact, SETD2 depletion resulted also in an increased γH2AX foci formation [32] SETD2 trimethylates histone-3 lysine-36 (H3K36me3) at sites of active transcription where the histone code reader LEDGF/p75 binds to it [13]. Furthermore, persistent γH2AX is an indicator for “oncogenic stress”, DNA damage driving genomic instability or malignant conversion [33]. Chromosomal instability and gene mutations in cancer cells are well described (reviewed in [34]) and are often caused by increased DNA DSBs with dysregulated DNA repair mechanisms. Since LEDGF/p75 is involved in the HDR-mediated DNA repair, its depletion might leave more DNA DSBs unrepaired. In cancer cells, spontaneous DNA damage foci formation (increase in γH2AX foci formation) caused by self-inflicting or endogenous DNA DSBs are often observed. We were able to show that in LEDGF KO cells, those endogenous DNA DSBs are significantly increased (Figure 4B). Moreover, LEDGF KO cells exhibited elevated γH2AX as well as BRCA1 foci, which are both known as DNA damage sensors [15,35]. Controversially, LEDGF KO cells could be continuously cultured (with decreased proliferation) and are not prone to go directly into complete cell death which suggests that LEDGF interacts with further signaling pathways.

Finally, increased phosphorylation of H2AX can be also caused in response to cell cycle progression during G2/M phase entry [36]. Furthermore, elevated γH2AX and G2/M arrest could also be indicators for senescence [37,38]. Therefore, a potential cell cycle arrest in LEDGF KO cells was investigated, especially because LEDGF knockdown has been previously shown to induce cell cycle arrest in the S/G2 phase and increased apoptosis in prostate cancer cells [10]. Nonetheless, in our study, cell cycle profiles of WT and LEDGF-depleted cells were similar without any sign of cell arrest in LEDGF KO cells (Appendix A). As we could not detect a cell cycle arrest but an increased chemosensitivity, we excluded the possibility that the γH2AX foci are senescence related. 

Since the persistent DNA fragmentation in LEDGF KO cells still allows proliferation, we investigated whether the persistently high γH2AX could be also related to an insufficient degradation of the γH2AX. An efficient and coordinated degradation of signaling molecules is necessary for an efficient DDR, which is also accompanied by a vast amount of proteasome-independent K63-linked ubiquitin modifications of the signaling molecules. The protein ubiquitylation plays a central role orchestrating the DDR, whereby K63 linkages promote protein recruiting and K48 linkages destabilize the protein (degradation) [39]. At the end of the signal transduction, many DDR-involved molecules e.g., γH2AX, are degraded by the proteasome [16,40,41]. Interestingly, it has been shown that the knockout of the nuclear proteasome activator PA28γ led to persistent γH2AX foci, even after signal termination [20]. These persistent γH2AX foci interfere with new DNA damage signals, which lead to an overall reduced DDR and thus to genetically unstable cells. Consequently, LEDGF-depleted cells were investigated for their PA28γ expression, and indeed, PA28γ protein levels were decreased. As demonstrated for PA28γ knockout cells, LEDGF depletion also resulted in a decreased PA28γ protein level. This might indicate that the sustained γH2AX foci and BRCA1 foci are also a result of the diminished degradation of these molecules. Moreover, the proteasome recycles ubiquitin by degrading proteins, which is required indirectly for the ubiquitin signaling in DDR, as it supplies the DNA repair ligases e.g., UBC13 with ubiquitin [42]. However, an effect of limited ubiquitin is very speculative and has been investigated in this study. Nonetheless, UBC13 has been previously suggested to ubiquitinate γH2AX molecules, which subsequently allows the binding of BRCA1 to initiate the degradation of γH2AX [43]. Therefore, the UBC13 expression was also verified in the LEDGF KO cells and as expected UBC13 protein levels were abrogated (Figure 5). A possible scenario how the LEDGF KO affects the HR-mediated DDR was illustrated in Figure 5G. 

In the past, LEDGF was only known to recruit the DDR protein CtIP; however, here, we show for the first time that LEDGF seems to have an indirect or direct effect on regulatory proteins of the DDR such as UBC13 and PA28γ, which are known to be involved, e.g., in the degradation of γH2AX.

## 4. Materials and Methods

### 4.1. Antibodies

The following antibodies were used in this study: Anti-RPA32 (9H8), anti-N-PSIP1, anti-BRCA1 (G-9), UBC13 (F-10) (Santa Cruz Biotechnology, Dallas, TX, USA), anti-C-LEDGF/p75 (Bethyl-Laboratories, Montgomery, TX, USA), anti-γH2AX (Cell Signaling, Danvers, MA, USA), anti-PA28γ (K5.6), α-Mouse-IgG-Atto 488 N (Sigma Aldrich, St. Louis, MO, USA), α-rabbit-IgG-Atto 647 (Dianova, Hamburg, Germany), anti-mouse-IgGκ BP-HRP (Santa Cruz Biotechnology, Dallas, TX, USA), and anti-rabbit HRP (Sigma Aldrich, St. Louis, MO, USA).

### 4.2. Cell Lines and Culture

Human epithelial type 2 (HEp-2) and human bone osteosarcoma epithelial cells (U2OS) wild-type (WT) cells as well as LEDGF knockout (KO), EGFP-LEDGF/p75 re-expressing (LEDGF/p75), and mEmarald_LEDGF/p52 re-expressing (LEDGF/p52) cells were grown up to 80% confluence in DMEM/Ham’s F12 supplemented with 10% FBS (Biowest, Nuaillé, France), 2 mM L-glutamine (Merck Millipore, Burlington, MA, USA), and penicillin/streptomycin (Merck Millipore, Burlington, MA, USA) in a humidified incubator at 37 °C and 5% CO_2_. WT and LEDGF/p75 cell lines were split in a 1:10 ratio, LEDGF KO cells were split in a 1:5 ratio.

### 4.3. Cloning

For the generation of LEDGF KO cells, sgRNA_DF70_E1 (AGATGAAAGGTTATCCCCAT, targeting exon 1 of *PSIP1* gene) was cloned into pSpCas9(BB)-2A-GFP (PX458; Addgene plasmid # 48138), kindly provided by Feng Zhang, Ph.D. [44]. Plasmids for EGFP-LEDGF/p75 overexpression (o/e) were created by cloning sgRNA_AAVS1 (CCAATCCTGTCCCTAG, targeting AAVS1 locus) into pSpCas9(BB)-2A-GFP and gBlock HDR fragment (attB-sites, EGFP-LEDGF/p75 coding sequence, Appendix A) into pAAVS1-P-CAG-DEST (Addgene plasmid #80490), kindly provided by Knut Woltjen, Ph.D. [45]. Plasmid for LEDGF/p52 o/e was created by cloning gBlock HDR fragment (attB-sites, mEmarald_LEDGF/p52 coding sequence, Appendix A) into pAAVS1-P-CAG-DEST as described in the GATEWAY Cloning Technology Instruction Manual.

### 4.4. Generation of LEDGF-Modified Cell Clones

HEp-2 WT and U2OS WT cells were seeded in 6-well plates (TH. Geyer, Renningen, Germany), incubated for 24 h, and subsequently transfected with px458_sgR_DFS70_E1 using LipofectamineTM 3000 according to the manufacturer’s instructions (Thermo Fisher Scientific, Waltham, MA, USA). For the EGFP-LEDGF/p75 o/e, WT and LEDGF KO cells were co-transfected with px458_sgRNA_AAVS1 and pAAVS1_CAG-EGFP-LEDGF/p75. The generation of mEmarald_LEDGF/p52 o/e cells was performed using px458_sgRNA_AAVS1 and pAAVS1_CAG-mEmarald_LEDGF/p52. Transfected cells were enriched by the EGFP selection of biomarkers via FACS using the S3e cell sorter (Bio-Rad). Briefly, cells were resuspended in 1× PBS supplemented with 0.5% FBS, while doublet cells were excluded by bivalent plotting of FSC and SSC, and positively transfected cells were sorted by GFP expression. Per 10 cm^2^ cell culture plate, a total of 1 × 10^3^ cells were seeded. Outgrown fluorescent, single cell colonies were picked after 7–10 days to establish LEDGF o/e cell lines. Subsequently, the cell clones were analyzed to verify the integration of the expression cassette at the AAVS1 locus was verified by PCR as described in Oceguera-Yanez [45], showing only heterozygous integrations. Subsequently, the expression of LEDGF was verified by Western blot, which exhibited different LEDGF expressions. Therefore, one cell clone expressing WT LEDGF and one LEDGF o/e was selected for the study.

### 4.5. Proliferation Analysis

To determine cell proliferation, cells were seeded at a density of 5 × 10^3^ cells/well in a 96-well plate (Th.Geyer, Renningen, Germany), and sulforhodamine B (SRB) assay was performed according to nature protocol [46].

### 4.6. Exposure to Etoposide

Cells were seeded at 5 × 10^3^ cells/well into 96-well plates (Th.Geyer, Renningen, Germany) and incubated for 24 h. Subsequently, cells were exposed to different levels of etoposide (between 2.5–20 µM).

### 4.7. Digital Image Analysis and Bioimage Informatics

Analysis of cell viability was performed using a fully automated multispectral fluorescence microscopy VideoScan platform [47,48]. Cells were stained with DAPI (2 µg/mL, Merck Millipore, Burlington, MA, USA) in 1× PBS (Biowest, Nuaillé, France) and incubated 15 min at 37 °C, 5% CO_2_, following VideoScan analysis with exposure time of 0.5–1 s. Cell count per image was measured via Blob detection as implemented in scikit-image (blob_log; v.0.17.2) [25] Images were further analyzed using bioimage informatics by Python 3.7 script as described elsewhere [25,49].

Cell morphology was analyzed via calculation of the roundness and eccentricity of each cell. For that, firstly, each cell with its corresponding nuclei was extracted as a set of points via thresholding (1.96% of maximum possible pixel intensity) and extraction of foreground pixels. Under-segmented areas were detected via the overlap of multiple nuclei with one area. Each region containing multiple nuclei was segmented via a seeded flood fill approach. Thereby, the centers of all corresponding nuclei were used as seed points for segmentation. Cell eccentricity was calculated via the second-order area moments. Roundess R was estimated as the normalized ratio between area A and perimeter P of the cell by the equation R = (4 ∗ π ∗ A)/P.

The morphology of cells was separated into normal and fibroblast-like via Density-Based Spatial Clustering of Applications with Noise (DBSCAN) clustering as implemented in the Python 3.7 package scikit-learn v. 0.23.1 (eps = 0.15, min_samples = 45). 

### 4.8. Scratch Assay

For measuring in vitro migration, 1 × 10^5^ cells/well were seeded in 48-well plates (Sarstedt, Nümbrecht, Germany) and incubated for 24 h. To stop proliferation, cells were treated with mitomycin C (10 µg/mL, abcr GmbH, Karlsruhe, Germany) for 2 h; then, they were washed and incubated for 22 h at 37 °C, 5% CO_2_. Circular scratches were created using a 100 µL pipette tip and measured after 0 h and 24 h. Scratch areas were analyzed using ImageJ-macro MRI_would_healing_tool (http://dev.mri.cnrs.fr/projects/imagej-macros/wiki/Wound_Healing_Tool, accessed on 14 December 2019).

### 4.9. Immunofluorescence

Cells were seeded at 5 × 10^3^ cells/well on 10-well slides (Hecht Assistant, Sondheim v. d. Rhön, Germany) and incubated for 24 h. For analysis, cells were fixed with 2% formaldehyde for 15 min at RT and permeabilized with 0.3% Triton X-100 (AppliChem, Darmstadt, Germany) while blocking with 5% BSA/PBS. Primary antibody was added and incubated at RT for 1 h. Slides were washed with 1× PBS and then incubated with secondary antibody and DAPI (2.5 mg/mL, 1:500) for 1 h in the dark at RT. Fluorophore photostability was increased by coating slides with mounting medium (Roti^®^-Mount FluorCare, Carl Roth GmbH, Karlsruhe, Germany). Analysis was performed using a confocal laser scanning microscope LSM 800 (Zeiss, Oberkochen, Germany). Foci formation (250–500 nuclei/image) was analyzed using NucDetect software, excluding mitotic cells (NucDetect 0.11.14, written in Python 3.7, available at https://pypi.org/project/NucDetect/, accessed on 25 March 2021).

### 4.10. Immunoblotting

Cells (1 × 10^6^) were harvested and resuspended in 50 µL 2× Lämmli-buffer, and protein content was determined using a Pierce^TM^ BCA Protein Assay Kit (Thermo Fisher Scientific, Waltham, MA, USA). Immunoblotting was performed using standard protocols [50]. Antibodies were diluted in 5% milk powder or 5% BSA (Carl Roth, Karlsruhe, Germany) in TBS/0.1% Tween-20 (AppliChem, Darmstadt, Germany). The band intensity was quantified by ImageJ software (1.53c 26).

### 4.11. Pulsed Field Gel Electrophoresis (PFGE)

Cells were seeded at 2 × 10^6^ cells/10 cm dish and incubated for 24 h. Cells were treated as indicated, harvested, cell pellets were resuspended in 1× PBS/peqGOLD agarose (VRW, Erlangen, Germany) (1:1) and used to cast inserts. Inserts were solidified at 4 °C and then incubated in ESP buffer (0.5 M EDTA (Carl Roth, Karlsruhe, Germany), 1% (*w/v*) N-laurylsarcosine (Carl Roth, Karlsruhe, Germany)) supplemented with 1.8 mg/mL proteinase K (≤30 mAnson U/mg, Carl Roth, Karlsruhe, Germany) at 56 °C for 18–20 h. Inserts were washed twice in 1 × TE buffer for 2 h at 4 °C, 10 rpm. Inserts were transferred to a 1.2% peqGOLD agarose (VWR, Darmstadt, Germany) gel in 0.5 × TBE buffer (44.5 mM Tris-borate and 1 mM EDTA (pH 8.3)) and run at 6 V/cm, 5–50 s switch time for 22 h at 14 °C in PFGE chamber (Bio-Rad Genepath Electrophoresis Gel Cell, Bio-Rad, Hercules, CA, USA).

### 4.12. Statistical Analysis

All data were statistically analyzed with the statistical computing language R v. 3.6 (R: The R Project for Statistical Computing, 2020 [51]). The Kolmogorov–Smirnov test was used for testing normal distribution. To control the α error inflation, the Bonferroni correction is applied or Tukey’s HSD test is used to test the differences between the mean values of the sample for significance. *p*-values less than 0.05 were considered significant. Experiments were conducted with at least three biological replicates. Data were further analyzed using RKWard v. 0.7.1z + 0.7.2 + devel2 [47] for the R statistical computing environment. Dose–response curves were fitted (95% confidence interval) with multiparametric functions (EXD3: Three-parameter exponential decay model; LL4: Four-parameter log-logistic model) from the drc package [52]. The optimal model was selected by using the AIC (Akaike information criterion) as criterion.

## Figures and Tables

**Figure 1 ijms-22-05866-f001:**
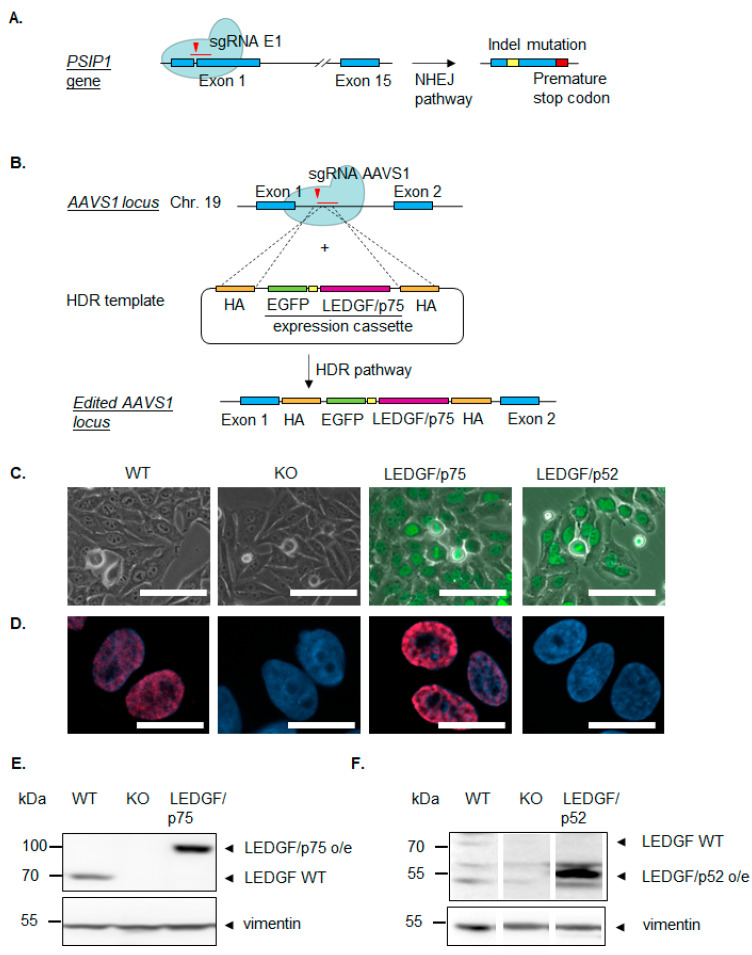
Verification of CRISPR/Cas9-mediated LEDGF knockout and LEDGF re-expression in HEp-2 cells. (**A**) Specific sgRNA for Exon 1 of LEDGF-coding gene *PSIP1* was designed to knockout (KO) LEDGF. The Cas9/sgRNA E1 complex induces double-strand breaks, which can be repaired by the cells through non-homologous end joining (NHEJ); however, NHEJ is error-prone, leading to indel mutations, which can cause premature stop codons. (**B**) LEDGF/p75 and LEDGF/p52 re-expressing cells were created by introducing a DNA DSB at a genomic safe-harbor locus (AAVS1) using an AAVS1-specific sgRNA. After the induction of a DSB, homology-directed repair (HDR) mediates the integration of the donor template containing the EGFP-LEDGF/p75 or a mEmarald_LEDGF/p52 expression cassette at the AAVS1 locus. Generated LEDGF knockout and re-expressing cells were verified by (**C**) fluorescence analysis with an excitation wavelength of 488 nm (scale bar = 100 µm), (**D**) indirect immunofluorescence (IF). Anti C-LEDGF antibody appear red due to conjugation to α-rabbit-IgG-Atto 647 secondary antibody, nuclei appear blue due to DAPI incorporation (scale bar = 20 µm). (**E**) Immunoblot using antibodies against C-terminal LEDGF and vimentin as loading control. (**F**) Immunoblot with antibodies against N-terminal LEDGF and vimentin as loading control.

**Figure 2 ijms-22-05866-f002:**
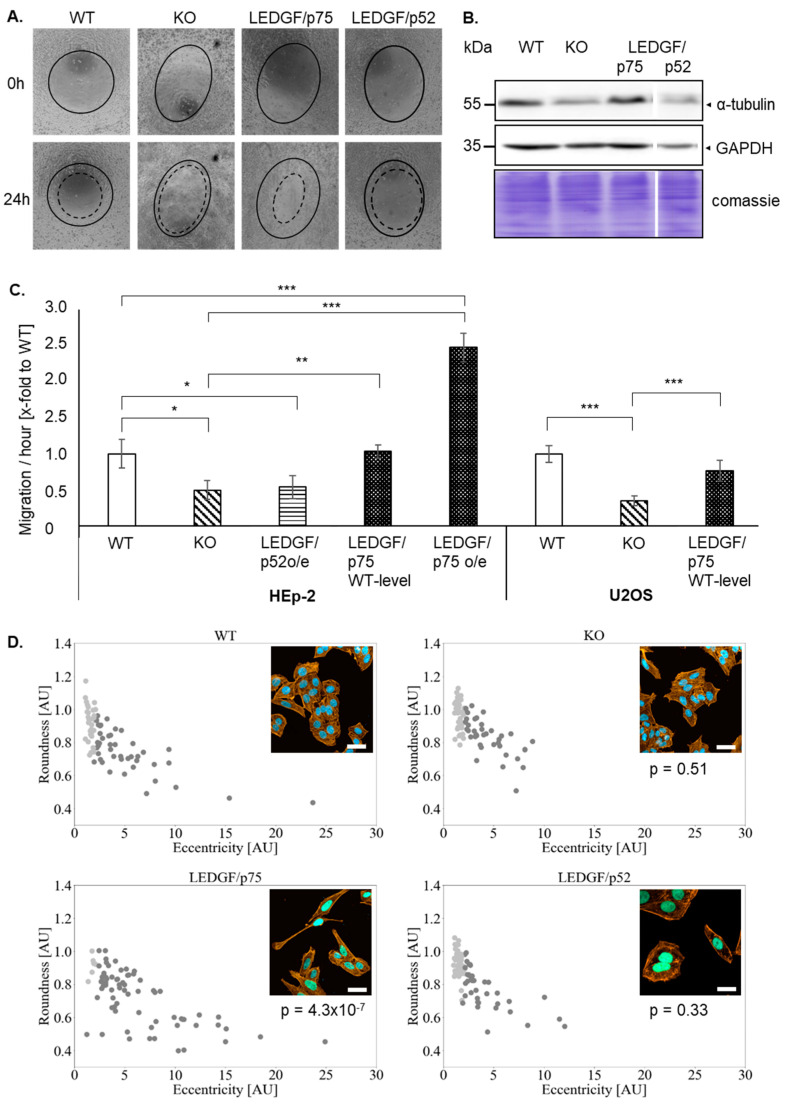
LEDGF influences cell migration and morphology. (**A**) Representative phase contrast image of HEp-2 WT, LEDGF K.O., LEDGF/p75, and LEDGF/p52 overexpressing cells, 0 h (black line) and 24 h (dashed line) after creating a circular scratch in a cell monolayer. Prior to the scratch, cells were incubated in 10 µg/mL mitomycin C to inhibit cell proliferation. (**B**) Immunoblot shows the level of α-tubulin in untreated cells. As loading control, gel was stained with Coomassie brilliant blue to visualize total protein amount (10 μg/lane), and GAPDH antibody was used. (**C**) The wound-healing capacity of the indicated HEp-2 and U2OS cell clones were analyzed using an ImageJ plugin MRI wound-healing tool after 24 h of the scratch induction and plotted as area/hour. (**D**) Representative confocal images of HEp-2 WT, LEDGF KO, LEDGF/p75, and LEDGF/p52 overexpressing cells were taken after incubation with Phalloidin-AlexaFluor555 (yellow). Chromatin appears blue due to DAPI incorporation, nuclei of LEDGF/p75 overexpressing cells appear green due to EGFP-tag of LEDGF/p75, scale bar = 20 µm. The images were used to perform a DBSCAN clustering correlating cell roundness and eccentricity in HEp-2 WT, LEDGF K.O., LEDGF/p75 and LEDGF/p52 overexpressing cells (*n* = 672, whereby only 75 cells are shown per condition). Cells with fibroblast-like morphology are shown in dark gray and cells with roundish morphology are shown in light gray. * *p* < 0.05, ** *p* < 0.01, *** *p* < 0.001.

**Figure 3 ijms-22-05866-f003:**
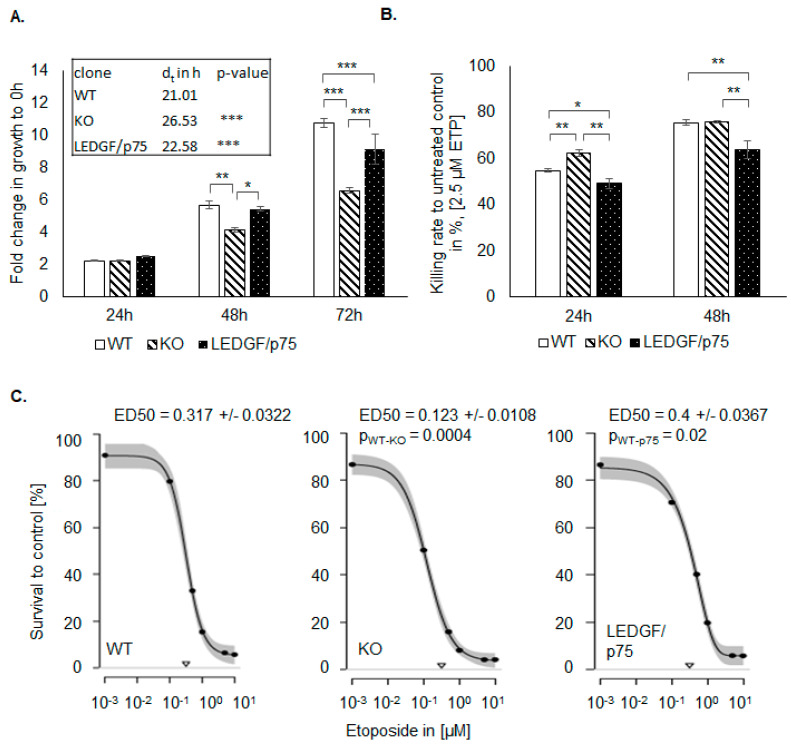
LEDGF affects cell proliferation and chemoresistance. (**A**) Proliferation of non-treated HEp-2 WT (passage 15–23), LEDGF KO (passage 6–13), and LEDGF/p75 re-expressing (passage 5–12) was determined for 0 h, 24 h, 48 h (pWT-KO = 0.006, pKO-p75 = 0.015, pp75-WT = 0.705), and 72 h (pWT-KO = 3 × 10^−6^, pKO-p75 = 7 × 10^−5^, pp75-WT = 0.03) by SRB assay. Doubling time (dt) was determined for the HEp-2 WT, LEDGF KO, and LEDGF/p75 re-expressing cell line (pWT-KO = 3.4 × 10^−5^, pp75-KO = 2.6 × 10^−4^). (**B**) After 24 h growth, indicated cell lines were treated with 1.25 µM etoposide for 24 h (pWT-KO = 0.010, pKO-p75 = 0.0004, pp75-WT = 0.022) and 48 h (pWT-KO = 0.044, pKO-p75 = 0.0002, pp75-WT = 0.0021), and the survival rate was determined in comparison to the untreated control. (**C**) Determination of ED50 value of HEp-2 WT, LEDGF KO, and LEDGF/p75 o/e cells after 48 h etoposide treatment, followed by 3 days of recovery. Dose–response curves were fitted (95% confidence interval) with multiparametric functions (EXD3: Three-parameter exponential decay model; LL4: Four-parameter log-logistic model, pWT-KO = 0.0004, pWT-p75 = 0.02). The triangles on the *x*-axis represent the dose of etoposide needed to reach ED_50_. * *p* < 0.05, ** *p* < 0.01, *** *p* < 0.001.

**Figure 4 ijms-22-05866-f004:**
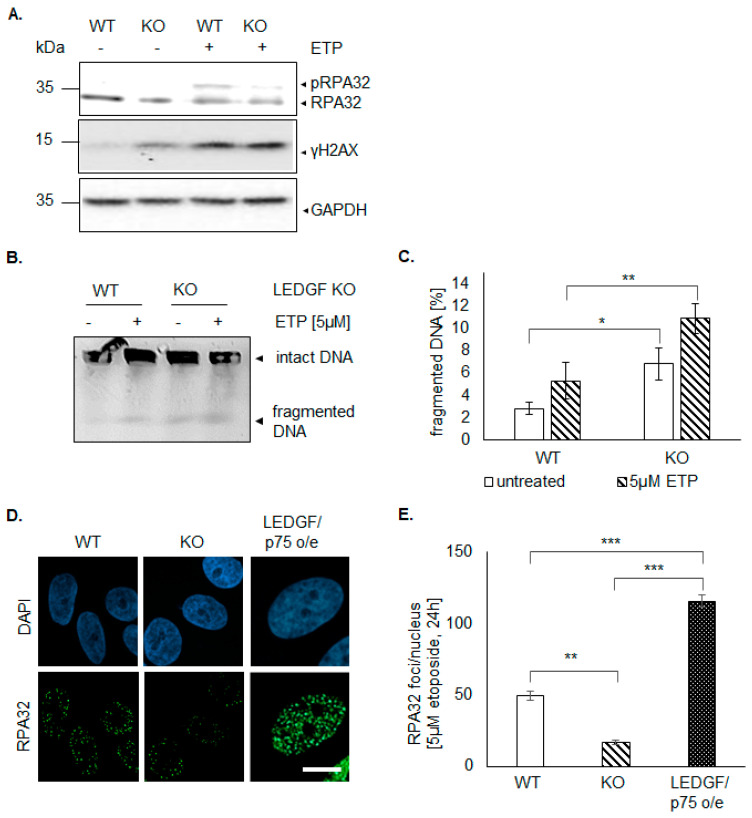
LEDGF necessary for CtIP-BRCA1-mediated homology-directed repair. (**A**) Immunoblots show levels of RPA32 (upper band phosphorylated RPA, pRPA), γH2AX, and GAPDH in untreated cells and cells after treatment with 5 µM etoposide for 6 h. (**B**) Pulsed field gel electrophoresis (PFGE) with 5 × 10^5^ cells/insert of HEp-2 WT and LEDGF KO cells. Cells were either not treated (“-”) or treated with 5 µM ETP for 6 h (“+”). (**C**) Quantitative analysis of fragmented DNA in HEp-2 WT and LEDGF KO cells as shown exemplary in Figure 4C. (*n* = 3). (**D**) Representative images of RPA32 foci are shown in indicated cells treated with 5 µM etoposide for 16 h. Cells were fixed with 2% formaldehyde and incubated anti-RPA32 (Santa Cruz Biotechnology, green). Chromatin appears blue due to DAPI incorporation. Scale bar: 10 µm. (**E**) Analysis of RPA32 foci (at least 100 cells were counted) was performed using NucDetect software, *n* = 3. * *p* < 0.05, ** *p* < 0.01, *** *p* < 0.001.

**Figure 5 ijms-22-05866-f005:**
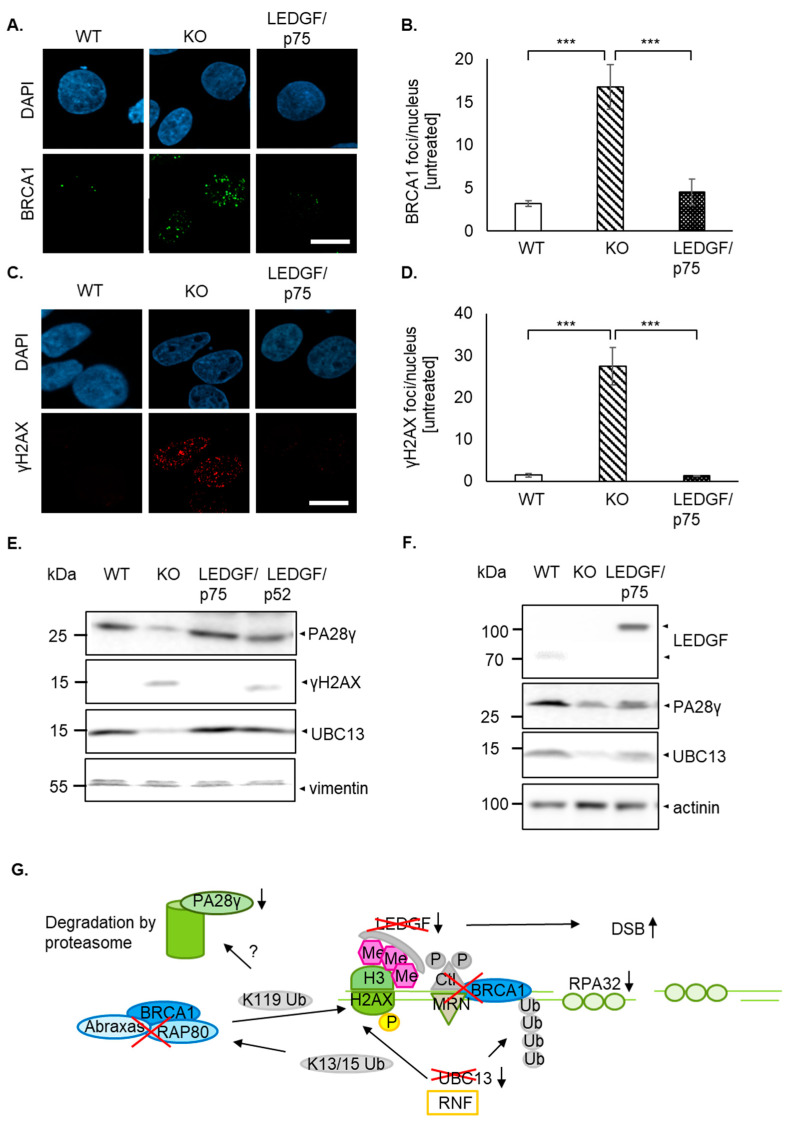
LEDGF depletion causes dysregulation of DNA damage response. (**A**,**C**) Representative confocal images of BRCA1 foci (**A**) and γH2AX foci (**C**) of untreated HEp-2 WT, LEDGF KO, and LEDGF/p75 re-expressing cells are shown after fixation with 2% formaldehyde and incubation with anti-γH2AX or anti-BRCA1. Chromatin appears blue due to DAPI incorporation, BRCA1 (green), and γH2AX (red). Scale bar: 10 µm. (**B**,**D**) Analysis of BRCA1 foci (pWT-KO = 0.0005, pKO-p75 = 0.0008, pWT-p75 = 0.8100) (**B**) and γH2AX foci (pWT-KO = 10^−7^, pKO-p75 = 10^−7^, pWT-p75 = 0.9934) (**D**) in HEp-2 WT, LEDGF KO, and LEDGF/p75 re-expressing cells using NucDetect software, *n* = 3. *** *p* < 0.00. (**E**) Untreated HEp-2 cell lines were harvested after 48 h and protein extracts were analyzed by immunoblotting using antibodies against PA28γ, γH2AX, UBC13, and vimentin as loading control. (**F**) Untreated U2OS cell lines were analyzed by immunoblotting. Vimentin was used as loading control. (**G**) Scheme of the HDR-mediated DDR signaling in LEDGF KO cells after DSB. LEDGF-dependent activation of CtlP is interrupted, allowing no complex formation with MRN and BRCA1 and subsequently no activation of RPA32 and later Rad51, resulting in ineffective DNA end resection. Due to UBC13 downregulation, K63-linked ubiquitination to lysine residues (K13/15) of γH2AX molecules are most likely missing. Without these ubiquitination signals, the BRCA1-A complex (containing also Rap80 and Abraxas) is unable to coordinate the release of γH2AX (ubiquitination at K119) from the chromatin. Consequently, degradation by the proteasome (in the nucleus PA28γ) is also impaired, leading to persistent γH2AX foci.

## Data Availability

The data that support the findings of this study are available from the corresponding author upon reasonable request.

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
