# Peer review of "LEDGF/p75 Is Required for an Efficient DNA Damage Response"

_ijms, 2021, doi:10.3390/ijms22115866_

Round 1

Reviewer 1 Report

Review of the manuscript entitled

‘LEDGF/p75 is required for an efficient DNA damage response’

by Victoria Liedtke, Christian Schröder, Dirk Roggenbuck, Romano Weiss, Ralf Stohwasser, Peter Schierack, Stefan Rödiger and Lysann Schenk

The authors in the presented manuscript entitled ‘LEDGF/p75 is required for an efficient DNA damage response’ present the role of lens epithelium derived growth factor splice variant of 75 kDa (LEDGF/p75) in DNA damage response. To explore the role of LEDGF, the authors have generated 3 cell models: a complete knock-out of LEDGF (KO) and re-expression of LEDGF/p75 or LEDGF/p52 using CRISPR/Cas9 technology. The study demonstrates the role of LEDGF/p75 in the regulation of proteins participating in homologous recombination.

In my opinion, the work should be published after minor corrections enlisted below.

  1. Line 42

‘SUMOylation of LEDGF/p75 [10] by sumo-specific pro-42 tease-1 regulates its binding in promoter regions of stress-related proteins’

includes a mistake. SENPs do not perform SUMOylation.

  1. Line 59/60

‘The LEDGF-de-pendent BRCA1-BARD-1 complex is a member of the E3-ubiquitin protein ligase family 60 [18].’

There is not information about LEDGF in [18].

  1. 1E

Please explain why do LEDGF/p75 protein localise at 100 kDa and not at 75 kDa band on WB in cells overexpressing LEDGF/p75.

Fig. 1E and 1F

Why two different antibodies were used for estimating long and short form of LEDGF? Is there complete knockout of LEDGF in Fig. 1F?

  1. 3

Is ED50 for PWT-KO correctly determined? What do triangles on X axis mean?

What do pWT-KO, pKO-p75 and pp75-WT mean?

  1. 5F

Which protein was used as reference actin or vimentin (please compare the figure with figure legends)? Why there is not equal expression of actin among samples?

  1. 5G

The figure is unclear and not compatible with figure legends.

  1. The publication cites in [4] is not adequate.

Author Response

  1. Line 42

‘SUMOylation of LEDGF/p75 [10] by sumo-specific pro-42 tease-1 regulates its binding in promoter regions of stress-related proteins’

includes a mistake. SENPs do not perform SUMOylation.

Response to Point 1: We thank the reviewer for this comment. SUMOylation is performed by SUMO proteins. We take out the part “by sumo-specific protease-1”

  1. Line 59/60

‘The LEDGF-de-pendent BRCA1-BARD-1 complex is a member of the E3-ubiquitin protein ligase family 60 [18].’

There is not information about LEDGF in [18].

Response to Point 2:  We call this complex LEDGF-dependent, because LEDGF is responsible for activating the acetylation of histone, 4 as written in the introduction part:“LEDGF/p75 recruits the histone acetyltransferase KAT5 to the chromatin which acetylates histone 4 (H4) at lysine K16 [13]. This acts as a switch between HDR and non-homologous end-joining (NHEJ). Upon H4 acetylation, the BRCA1-BARD1 complex can bind to DNA supporting HDR”

  1. 1E

Please explain why do LEDGF/p75 protein localise at 100 kDa and not at 75 kDa band on WB in cells overexpressing LEDGF/p75.

Response to Point 3:  As stated in 1B, LEDGF/p75 re-expressing cells were created by introducing a DNA DSB at genomic safe-harbour locus (AAVS1) using an AAVS1-specific sgRNA. After the induction of a DSB, homology-directed repair (HDR) mediates the integration of the donor template containing the EGFP-LEDGF/p75. Due to this EGFP-tag LEDGF/p75 localises at 100kDa, whereas WT LEDGF isdetected at 75kDa.

Fig. 1E and 1F

Why two different antibodies were used for estimating long and short form of LEDGF? Is there complete knockout of LEDGF in Fig. 1F?

Response to Point 3:  Anti-C-LEDGF/p75 (Bethyl-Laboratories, Texas, USA) binds  LEDGF (from amino acid 430 until C-term) and was used to detect the long of LEDGF, while the short form of LEDGF remains undetected due to the missing C-terminal part. N-PSIP1 (Santa Cruz Biotechnology, Texas, USA) binds to the N-terminus of LEDGF and therefore detects both forms of LEDGF. The LEDGF KO clone used in this work (and shown in figure 1E, 1F) is a complete LEDGF KO, verified by sequencing. Faint bands in figure 1F lane 2 can therefore be regarded as background or non-specific binding. 

  1. 3

Is ED50 for PWT-KO correctly determined? What do triangles on X axis mean?

What do pWT-KO, pKO-p75 and pp75-WT mean?

Response to Point 4:  ED50 for WT-KO is correctly determined and shows that LEDGF KO has a significant impact on the cell recovery after etoposide treatment.  Triangles on x-axis represent the concentration of etoposide needed for ED50. pWT-KO, pKO-p75 and pp75-WT represent p-values obtained by Kruskal-Wallis test to allow conclusions between the different cell lines. 

  1. 5F

Which protein was used as reference actin or vimentin (please compare the figure with figure legends)? Why there is not equal expression of actin among samples?

Response to Point 5:  We thank the reviewer for pointing this out and have adjusted the figure legend.

  1. 5G

The figure is unclear and not compatible with figure legends.

Response to Point 6:  We adjusted the figure legend of 5G and hope we clarified everything.

  1. The publication cites in [4] is not adequate.

Response to Point 7:  Thanks for pointing this out, this citation was added by mistake. We replaced the citation with the actual citation: Basu et al, 2012 and Singh et al., 2017

Reviewer 2 Report

Liedtke et.al., in their article, "LEDGF/p75 is required for an efficient DNA damage response" present groundbreaking data on the biological response of p75, one key DDR molecule, in the homology directed DNA repair pathway. The authors, build a useful stable LEDGF KO cell line, which is then compounded by either the expression of p75, or p52. Using this powerful technology, they address key questions on the biology of p75 in the DNA damage response. Their studies indicate that p75 exclusively aid in cancer phenotypes by enabling proliferation, migration, resistance to cancer drugs such as etoposide. In addition, p75 also seems to play a role in engaging the downstream molecules to facilitate the DDR response through RPA, BRCA1, and phosphorylated H2AX. Strikingly, this DDR protein, p75, also sustains the ubiquitination signals, thereby bringing in a new insight as to how to think about the DDR pathway from a p75 perspective. 

A few minor questions that did arise, were the choice of the DNA damaging agent: why was etoposide used? and did the authors confirm a similar phenotype to IR (ionizing radiation)? A second question that comes to mind is whether the authors observed other reported phenotypes in the DDR context--mitotic, cell cycle, translesion synthesis..?

The paper is well written, very nicely laid out and the authors are congratulated on a job well done..

Author Response

1. A few minor questions that did arise, were the choice of the DNA damaging agent: why was etoposide used?

Response to Point 1:  As etoposide is widely used to treat cancers such as lung cancer, lymphoma or non-lymphocytic leukemia, it is an ideal reagent to study DNA double-strand breaks with clinical relevance. In contrast to platinum drugs, it does not bind covalently to DNA but forms a complex with topoisomerase II to interrupt DNA replication. Irradiation would also have been a good way to investigate DNA damage, unfortunately there was no radiation facility available. 

2. and did the authors confirm a similar phenotype to IR (ionizing radiation)?

Response to Point 2:  IR would have been an obvious choice as it has been used widely to investigate DNA damage repair, unfortunately our institute has no ionizing radiation facility.  

3. A second question that comes to mind is whether the authors observed other reported phenotypes in the DDR context--mitotic, cell cycle, translesion synthesis..?

Response to Point 3:  Since HR occurs in S-phase, we suspected a prolonged S-phase but cell cycle analysis reveals no change in cell cycle progression between WT, KO and LEDGF/p75. Therefore, we based our arguments on the observed increased etoposide sensitivity, the decreased migration and proliferation, and the accumulation of yH2AX foci in LEDGF KO cells.

Reviewer 3 Report

This article focuses on understanding the effects of LEDGF/p75 knockout (KO) on cell growth and the DNA damage response (DDR). LEDGF/p75 is a transcriptional regulator that binds to chromatin, has anti-apoptotic activity and is overexpressed in several cancers. However, the mechanism by which excess LEDGF/p75 promotes cancer and functions is still not fully characterized. In this study the authors set out to determine the effects of LEDGF KO in human cells, using CRISPR-Cas9 technology. The authors make two main conclusions. First that LEDGF/p75 is important for migration and cell growth but not cell cycle progression. Second “participation of LEDGF/p75 in the ubiquitin-dependent regulation of DDR signaling molecules”. From this they develop a complex model whereby loss of LEDGF affects BRCA1 ubiquitination and its ability to complex with MRN-CtIP and Abraxas-RAP80, which leads to unresolved DSBs that persist over multiple divisions in LEDGF KO cells. Unfortunately, in this reviewer’s opinion, while the human KO lines will be important to assess the role of LEDGF, the data presented here do not support the authors major conclusions and proposed model. Additionally, many of these experiments were already performed with siRNA knockdown of LEDGF in HeLa and U2OS cells and LEDGF KO MEFs (Daugaard et al. Nat Struct Mol Biol, 2012). This study found similar findings and performed much more mechanistic analysis of LEDGF in DSB repair. Therefore, it is not clear what additional insights are gained from these studies. While there is some interesting data, additional experiments would need to be performed as well as the quality and rigor of the data improved to warrant publication. As presented, I do not support publication of the manuscript at this time. Below are specific concerns that the authors should consider to improve the manuscript.

Major:

  1. Detailed analysis of the growth defects, apoptosis and the DDR in the absence of DNA damaging agents would be a more novel aspect of this study compared to the previous one (Daugaard et al. Nat Struct Mol Biol, 2012). However, these phenotypes are not well characterized in the study. The authors state in the conclusions that “LDGF cells could be cultured …and are not prone to go directly into apoptosis”. This could be an interesting finding but I am unclear what data supports this claim. They have not directly measured apoptosis, apoptotic markers or done growth curve analysis to directly test this. This is particularly important since LEDGF/p75 is considered an anti-apoptotic factor. Senescence is another possibility to explain the decreased growth but was not investigated. Unless the cells are arrested or moving very slowly through the cell cycle, which is not seen in the flow data, it is not clear to this reviewer why there is decreased proliferation. Furthermore, the authors claim that the DDR is being affected in the KO cells but do not assess major activators of the DDR, such as ATM/CHK2, ATR/CHK1, DNA-PK.
  2. The authors state that “ubiquitin-dependent regulation of the of DDR signaling molecules” is affected in LEDGF KO cells. However, this is based solely on a decrease in UBC13 and PA28g and not any direct evidence. Decreased ubiquitination of BRCA1 (or other repair/DDR factors) or changes in proteosome degradation of such factors was not shown. Changes in CtIP/MRN localization to DSBs, decreased BRCA1 interaction with Abraxis-Rap80 should also be looked at.
  3. “RPA32 foci formation” should not be “used as a surrogate to detect active HR”, as stated in the manuscript. I am also confused why RPA foci should increase in the LEDGF KO cells, since the previous study found that resection was inhibited and RPA/RAD51 foci decreased following LEDGF knockdown due to loss of CtIP localization (Daugaard et al. Nat Struct Mol Biol, 2012). RPA is also found at stalled replication forks or other types of DNA damage so it possible that the increase is due defective replication or slower S-phase progression. Co-localization studies, for example with gH2AX and phosphorylated RPA, BRCA1, RAD51 or other repair factors, and the exclusion of replicating cells (e.g. EdU positive cells) could determine whether these are persistent DSBs. EdU or BrdU staining combined with flow cytometry, or DNA fiber analysis, could also be used to show that replication is not affected. Regarding the RPA foci, it is also not shown whether these foci are only present when cells are treated with etoposide or in untreated cells as well. Finally, the RPA32 graph does not provide information on how many cells are RPA positive. A dot plot or similar graph would show whether all, most or some of the cells have RPA32 foci.
  4. Figure 4B is used to show that DSBs increase in the LEDGF KO cells even without etoposide treatment. They claim that this shows “massive DNA fragmentation” but the levels do not look that large, at least as presented. Quantification should be included to compare the WT and KOs. Alternatively, they could show increased DSBs by another technique like comet assay. Confirmation of these results in the U2OS cell line would also increase confidence in these results. This needs to be provided, especially since it is critical to their claim that DSBs are increased and persist in the KO cells over time. Another explanation could be that LEDGF KO cells are more prone to DSBs/DNA damage leading to cell death versus the DSBs persisting over time.
  5. The U2OS LEDGF KO line has not been as well characterized as the HEp-2 cells. A western blot has not been shown for these KO cells or what has been deleted. The only data shown is the migration assay and the BRCA1 and gH2AX foci. What about cell growth, cell cycle and DSBs?
  6. The lanes in Figures 1F and S4 are separate from each other. Are these from different blots or have they been cropped from the same blot? Also, the blot in Figure 1F is not very convincing. The LEDGF WT signal is weak in the control and looks like there may be a slight in the KO and the exposures appear to be different in between lanes. Obtaining a gel with all the samples run together/side-by-side and darker exposures, or at least the full blots and the lanes used should be shown.

Minor:

  1. In Figure 1A, it is not clear why exon 15 is crossed out? Wouldn’t the entire gene not be expressed past the premature stop codon in exon 1?
  2. What the stars (p-values) in the graphs represent should be stated in the figure legends.
  3. Could the authors explain in the results or discussion the significance of the migration and cell morphology studies presented in Figure 2?
  4. Can you explain the difference between the p75 WT-level and p75 o/e cells and how they were generated?
  5. Why is only etoposide used in this study and not other DNA damaging agents?
  6. Manuscript would benefit from proofreading to correct a number of grammatical errors throughout.

Author Response

Major:

1. Detailed analysis of the growth defects, apoptosis and the DDR in the absence of DNA damaging agents would be a more novel aspect of this study compared to the previous one (Daugaard et al. Nat Struct Mol Biol, 2012). However, these phenotypes are not well characterized in the study. The authors state in the conclusions that “LDGF cells could be cultured and are not prone to go directly into apoptosis”. This could be an interesting finding but I am unclear what data supports this claim. They have not directly measured apoptosis, apoptotic markers or done growth curve analysis to directly test this. This is particularly important since LEDGF/p75 is considered an anti-apoptotic factor. Senescence is another possibility to explain the decreased growth but was not investigated. Unless the cells are arrested or moving very slowly through the cell cycle, which is not seen in the flow data, it is not clear to this reviewer why there is decreased proliferation. Furthermore, the authors claim that the DDR is being affected in the KO cells but do not assess major activators of the DDR, such as ATM/CHK2, ATR/CHK1, DNA-PK.

Response to Point 1: It has been previously described that LEDGF knockdown results in reduced proliferation (Basu et al., 2012). Nonetheless, CRISPR-mediated LEDGF KO might behave differently than siRNA-induced LEDGF KO. However, as described in Figure 3 A, we could detect growth inhibition upon LEDGF KO even without the influence of DNA damaging agents. In the presented study, we have analysed growth curves over a period of 72h, the data show a significant growth reduction after knockout of LEDGF which is also reflected by a decreased doubling time. Using the microscope, we did not detect any obvious apoptotic cells in comparison to the WT cells. Moreover, we were able to passage the cells for 20-30 passages with no change in growth or death behavior.

Concerning the analysis of apoptosis levels, we agree with the reviewer that more analysis would be beneficial. At the moment, another paper is in progress looking at the relationship between LEDGF knockout and apoptosis developing an assay in parallel.

We agree with the reviewer that we have not explicitly analysed the activities of ATM/CHK2, ATR/CHK1 or DNA-PK. However, our experiments include the indirect measurement of DNA PK activity through the phosphorylation of histone 2AX and RPA32.

Of course yH2AX foci might be also an indicator for the senescence but no further obvious characteristic was seen in the analysed cells. The molecular characteristics of senescent cells can have various characteristics: an upregulation of cell-cycle inhibitors e.g.  p21, positive staining of senescence-associated β-galactosidase (SA-β-gal), formation of senescence-associated heterochromatin foci (SAHF) and/or the induction of senescence-associated DNA damage e.g. yH2AX [34, 35]. Nonetheless, senescent cells should be non-dividing and are resistant to stress stimuli e.g. etoposide but in fact, LEDGF KO cells are more sensitive to etoposide. In addition, we don’t see any difference of chromatin staining with DAPI showing any characteristic foci formation. Moreover, we did not see any G2/M arrest in the LEDGF KOs.

2. The authors state that “ubiquitin-dependent regulation of the of DDR signaling molecules” is affected in LEDGF KO cells. However, this is based solely on a decrease in UBC13 and PA28g and not any direct evidence. Decreased ubiquitination of BRCA1 (or other repair/DDR factors) or changes in proteosome degradation of such factors was not shown. Changes in CtIP/MRN localization to DSBs, decreased BRCA1 interaction with Abraxis-Rap80 should also be looked at. 

Response to Point 2: We agree with the reviewer, that we did not show direct evidence, but we use UBC13 and PA28gamma as indirect evidence for LEDGF’s involvement in ubiquitin-dependent regulation of DDR signaling. As shown by Levy-Barda in 2011, PA28gamma knockout leads to a delay in DNA DSB repair. They also show that loss of PA28gamma leads to persistent chromatin compounds, which matches our results of persistent yH2AX foci in LEDGF KO cells. Changes in CtIP/MRN localisation or BRCA1 interaction with Abraxix-RAp80 are a good argument but were not feasible to investigate in the given time.  

3. “RPA32 foci formation” should not be “used as a surrogate to detect active HR”, as stated in the manuscript. I am also confused why RPA foci should increase in the LEDGF KO cells, since the previous study found that resection was inhibited and RPA/RAD51 foci decreased following LEDGF knockdown due to loss of CtIP localization (Daugaard et al. Nat Struct Mol Biol, 2012). RPA is also found at stalled replication forks or other types of DNA damage so it possible that the increase is due defective replication or slower S-phase progression. Co-localization studies, for example with gH2AX and phosphorylated RPA, BRCA1, RAD51 or other repair factors, and the exclusion of replicating cells (e.g. EdU positive cells) could determine whether these are persistent DSBs. EdU or BrdU staining combined with flow cytometry, or DNA fiber analysis, could also be used to show that replication is not affected. Regarding the RPA foci, it is also not shown whether these foci are only present when cells are treated with etoposide or in untreated cells as well. Finally, the RPA32 graph does not provide information on how many cells are RPA positive. A dot plot or similar graph would show whether all, most or some of the cells have RPA32 foci.[1]

Response to Point 3:  Initially, we were interested in LEDGF because it has been previously suggested to be involved in HR [20] . In fact, as shown in figure 4,  RPA32 foci are significantly reduced in LEDGF KO cells upon etoposide treatment. In untreated cells, basically no RPA32 foci were detectable (data not shown) however, we have observed increased yH2AX foci and DNA fragmentation combined with reduced growth upon LEDGF KO (Fig. 3 and 5). All these indicates a defective DNA damage. In addition, LEDGF KO showed an increased sensitivity towards etoposide which might indicate, as the reviewer suggested, a slower S-phase progression since HR only occurs in S-phase and defective HR would most likely lead to a prolonged S-phase as DNA damage is insufficiently repaired. In addition, cells would be more prone to die in S-phase rather than in G2-phase. In this respect, the reviewer is completely right, a BrdU staining would bring more clarity. Unfortunately, the given revision time was too short to allow any additional experiments.
Additionally, we changed the description of the methods section immunofluorescence to clarify that mitotic cells are excluded from the measurement of foci. Furthermore, supplement 7 was added to show how many cells are positive for the investigated foci formation and how many foci in each nucleus were detected. 

4. Figure 4B is used to show that DSBs increase in the LEDGF KO cells even without etoposide treatment. They claim that this shows “massive DNA fragmentation” but the levels do not look that large, at least as presented. Quantification should be included to compare the WT and KOs. Alternatively, they could show increased DSBs by another technique like comet assay. Confirmation of these results in the U2OS cell line would also increase confidence in these results. This needs to be provided, especially since it is critical to their claim that DSBs are increased and persist in the KO cells over time. Another explanation could be that LEDGF KO cells are more prone to DSBs/DNA damage leading to cell death versus the DSBs persisting over time.

Response to Point 4:  We agree that "massive DNA fragmentation" was a bit of an overstatement, so we changed it to "significantly increased amount of DNA fragmentation" and added statistical analysis of three independent PFGE experiments (Fig 4C). Confirming these results in U2OS cells is part of the next paper, where the effect of LEDGF KO to cell growth and apoptosis is analysed.

5. The U2OS LEDGF KO line has not been as well characterized as the HEp-2 cells. A western blot has not been shown for these KO cells or what has been deleted[2] . The only data shown is the migration assay and the BRCA1 and gH2AX foci. What about cell growth, cell cycle and DSBs? 

Response to Point 5:  In supplement 3, a more detailed characterisation of U2OS LEDGF KO cells was added including LEDGF protein levels of various cell clones as well as sequencing analysis of the targeted locus and  some potential off-target regions. Furthermore, we also performed growth and toxicity assays with U2OS cells which showed similar results as in HEp-2 cells (data not shown).

6. The lanes in Figures 1F and S4 are separate from each other. Are these from different blots or have they been cropped from the same blot? Also, the blot in Figure 1F is not very convincing. The LEDGF WT signal is weak in the control and looks like there may be a slight in the KO and the exposures appear to be different in between lanes[3] . Obtaining a gel with all the samples run together/side-by-side and darker exposures, or at least the full blots and the lanes used should be shown.

Response to Point 6:  The original uncut WB data for this paper show that the lanes of 1F and S4 are located on the same blot, but are not directly next to each other. For better clarity, the individual lanes are shown separately from each other. In addition, a different WB was chosen in which the expression is more clearly visible

Minor:

1.In Figure 1A, it is not clear why exon 15 is crossed out? Wouldn’t the entire gene not be expressed past the premature stop codon in exon 1?

Response to Point 1:   Figure 1 was revised. The cross over exon 15 was misleading, so we revised the figure to clarify that no transcription occurs after the premature stop codon.

2. What the stars (p-values) in [5] the graphs represent should be stated in the figure legends.

Response to Point 2: Thanks for pointing this out, we have added the p-values to the legends of each figure.

3. Could the authors explain in the results or discussion the significance of the migration and cell morphology studies presented in Figure 2?

Response to Point 3: Previously, LEDGF has been shown to be involved in migration (Basu et al., 2017). Thus, we wanted to verify the same effect upon LEDGF knockout instead of knockdown. Moreover, migration is critical for cancer progression. After the initial experiments, we observed the changed morphology and investigated this feature further as it has not been described previously. Our findings suggest that LEDGF overexpression leads to a more aggressive and more invasive cancer phenotype.

4. Can you explain the difference between the p75 WT-level and p75 o/e cells and how they were generated?

Response to Point 4: Both cell lines were generated in the same manner. LEDGF KO cells were co-transfected with a pSpCas9(BB)-2A-GFP plasmid containing sgRNA_AAVS1 (targeting AAVS1 locus) and a pAAVS1-P-CAG-DEST plasmid containing a gBlock HDR fragment. Transfected cells were enriched by GFP expression and adult fluorescent single cell colonies were harvested after 7-10 days. Western blot and immunofluorescence analysis showed cells with moderate LEDGF/p75 expression, similar to WT cells, so we named them "p75 WT level" and other clones showed an upregulated level of LEDGF/p75,  named  "p75 o/e". In order to show that the level of LEDGF expression is important for their migratory capacity, we used different expression clones.

5. Why is only etoposide used in this study and not other DNA damaging agents?

Response to Point 5: As etoposide is widely used to treat various cancers such as lung cancer, lymphoma or non-lymphocytic leukemia, it is an ideal chemotherapeutic agent to study DNA double-strand breaks with clinical relevance. In contrast to platinum drugs, it does not bind covalently to DNA but forms a complex with topoisomerase II to interrupt DNA replication. Irradiation would also have been a good way to investigate DNA damage, unfortunately our institute has no radiation facility..

6. Manuscript would benefit from proofreading to correct a number of grammatical errors throughout.

Response to Point 6: We are not native English speakers and therefore we are always grateful to get some advice to improve our written English. Therefore, we  asked a native English speaker to check the  manuscript and some things have been improved . If there are still any grammatical inconsistencies that we should change (could you please give examples if you are not happy with stylistic things), we are happy to revise them.

Round 2

Reviewer 3 Report

While I appreciate the authors responses and addressing some of the concerns, no new data has been added that directly address other concerns raised about what is causing the growth defect, a detailed analysis of the cell cycle checkpoint and direct evidence for their model. From their comments, it appears that this may be due to a time constraint placed on them to resubmit the revised manuscript. However, without additional data, my opinion remains that their conclusions are still not fully supported by the current results and I still cannot recommend the article for publication at this time. While I do not feel that all of the concerns raised need to be addressed, at least some new data is needed to bolster support for the conclusions and model. Alternatively, the authors could tone down their conclusions.

As the other reviewers find the article acceptable, if the publication is to still go ahead, I recommend that the authors at least provide the source data for the Western blots (see major, point #6), provide what pRPA32 antibody used in the methods (i.e. S4/8, S33) and explain the p75 WT vs p75 o/e in the methods section (see minor point #4).

Author Response

While I appreciate the authors responses and addressing some of the concerns, no new data has been added that directly address other concerns raised about what is causing the growth defect, a detailed analysis of the cell cycle checkpoint and direct evidence for their model. From their comments, it appears that this may be due to a time constraint placed on them to resubmit the revised manuscript. However, without additional data, my opinion remains that their conclusions are still not fully supported by the current results and I still cannot recommend the article for publication at this time. While I do not feel that all of the concerns raised need to be addressed, at least some new data is needed to bolster support for the conclusions and model. Alternatively, the authors could tone down their conclusions.

As the other reviewers find the article acceptable, if the publication is to still go ahead, I recommend that the authors at least provide the source data for the Western blots (see major, point #6), provide what pRPA32 antibody used in the methods (i.e. S4/8, S33) and explain the p75 WT vs p75 o/e in the methods section (see minor point #4).

Response 1: We thank the reviewer for this decision. All original data were uploaded upon submission and are available to the journal. Alternatively, we can offer to show the original data for the Western blot in the supplements. 

Response 2: In fact, we used a “total RPA32” antibody (9H8) as described in the method section. Phosphorylated RPA32 migrates slower through the acrylamid gel and can be discriminated from the non-phosphorylated RPA32.The description for pRPA detection was  misleading therefore we  amended the description in the results section and in the figure legend of figure 4.  Additionally, this was confirmed by immunofluorescence. There, RPA shows up as foci only after phosphorylation.

Response 3: Both LEDGF_p75 (WT level) and LEDGF_p75 o/e were generated in the same way. We followed the paper of Oceguera-Yanez (2016) [44] and thus caused a directional integration of the donor template EGFP-LEDGF/p75 into the AAVS1 locus. The analysis of both cell clones by sequencing and PCR (data not shown)  detected an AAVSI WT allele while the EGFP-LEDGF/p75 template was integrated on the other allele (heterozygous genotype). Nonetheless, the different heterozygous clones exhibited different protein expression as verified by  Western blot (supplement 4) which expression variability has already been described previously (doi: 10.12688/f1000research.19894.2). The different expression level is maybe a result of different regulations of the transgene expression through the clone-specific effect on e.g. the constitutive promoter EF1alpha. For our experiments, the different protein levels of LEDGF were important to analyze the direct effects of o/e LEDGF versus LEDGF KO.